# The Effect of Alkali Metals (Li, Na, and K) on Ni/CaO Dual-Functional Materials for Integrated CO_2_ Capture and Hydrogenation

**DOI:** 10.3390/ma16155430

**Published:** 2023-08-02

**Authors:** Yong Hu, Qian Xu, Yao Sheng, Xueguang Wang, Hongwei Cheng, Xingli Zou, Xionggang Lu

**Affiliations:** State Key Laboratory of Advanced Special Steel, School of Materials Science and Engineering, Shanghai University, 99 Shangda Road, BaoShan District, Shanghai 200444, China; huyongsh@163.com (Y.H.); shengyao@shu.edu.cn (Y.S.); hwcheng@shu.edu.cn (H.C.); xlzou@shu.edu.cn (X.Z.); luxg@shu.edu.cn (X.L.)

**Keywords:** alkali metal, Ni/CaO, dual-functional material, CO_2_ capture and hydrogenation, reverse water–gas shift

## Abstract

Ni/CaO, a low-cost dual-functional material (DFM), has been widely studied for integrated CO_2_ capture and hydrogenation. The core of this dual-functional material should possess both good CO_2_ capture–conversion performance and structural stability. Here, we synthesized Ni/CaO DFMs modified with alkali metals (Na, K, and Li) through a combination of precipitation and combustion methods. It was found that Na-modified Ni/CaO (Na-Ni/CaO) DFM offered stable CO_2_ capture–conversion activity over 20 cycles, with a high CO_2_ capture capacity of 10.8 mmol/g and a high CO_2_ conversion rate of 60.5% at the same temperature of 650 °C. The enhanced CO_2_ capture capacity was attributed to the improved surface basicity of Na-Ni/CaO. In addition, the incorporation of Na into DFMs had a favorable effect on the formation of double salts, which shorten the CO_2_ capture and release process and promoted DFM stability by hindering their aggregation and the sintering of DFMs.

## 1. Introduction

The excessive emissions of anthropogenic CO_2_ into the atmosphere are recognized as a primary driver of global warming and climate change [1,2,3]. To combat this issue, collaborative efforts from researchers and governments are required to curtail CO_2_ emissions. Carbon capture and storage (CCS) have been considered the primary strategy to achieve emission reduction targets, but its high cost and significant energy consumption in CO_2_ purification, compression, and transportation limit its widespread application [4,5,6,7]. An alternative approach, known as CO_2_ capture and utilization (CCU), aims to convert the captured CO_2_ into value-added fuels and chemicals, offering a promising solution to the CO_2_ challenge [8,9,10]. Integrated CO_2_ capture and utilization (ICCU) technology, which combines capture and conversion in a single column under the same temperature, has garnered increasing interest due to its potential to reduce capital and operating costs while saving energy [11,12,13].

Generally, the successful implementation of ICCU relies on the development of dual-functional materials (DFMs) capable of serving as both CO_2_ adsorbents and catalytic components for CO_2_ conversion. DFMs cyclically capture CO_2_ from combustion flue gas with a CO_2_ concentration of 4–14 vol%, subsequently converting the adsorbed CO_2_ into valuable products, such as synthetic fuels or other chemicals, while regenerating the adsorbent. CO_2_ adsorbents are typically based on alkaline or alkaline–earth compounds, as they react with acidic CO_2_ to form corresponding carbonates. Among these materials, CaO-based composites are extensively studied in ICCU processes, due to their low cost, high availability (e.g., limestone), and excellent theoretical CO_2_ capacity (17.8 mmol/gCaO) [14,15]. On the other hand, catalysts such as Pt [16], Ni [17], or Fe [18] are widely used to catalyze the C1 chemistry reactions, including methanation, reverse water–gas shift (RWGS), and dry reforming of methane (DRM) in ICCU processes for CO_2_ conversion. As a result, Ni/CaO DFMs are extensively employed in ICCU technology due to their optimal cost-to-activity ratio [19,20,21,22,23,24,25,26,27]. However, the development of Ni/CaO DFMs with both high CO_2_ capture capacity and efficient catalytic activity remains a significant challenge since most CO_2_ conversion reactions are conducted outside at a high temperature, but CaO does not maintain its stability during carbonation–calcination cycles at high temperatures, given the relatively low Tammann temperature of CaCO_3_ (533 °C). Numerous efforts have been dedicated to resolving this sintering issue. In previous studies, we successfully applied a three-dimensional Ni/CaO network composed of mesopores and macropores in a stable high-temperature ICCU process thanks to its unique porosity structure [28,29]. Recently, the doping approach of DFMs with alkali metals, such as Li, Na, K, and Cs, has been considered a promising strategy to promote the ICCU process. Previous research has developed Ca adsorbents and investigated the promoting effect of alkali metals. Ahmed et al. conducted a study on a series of double salts, which included K- and Na-promoted Ca adsorbents, specifically designed for CO_2_ capture at 650 °C. The results of their research showed that the K−Ca double salt displayed an exceptional CO_2_ adsorption capacity of 10.7 mmol/g [30]. The K−Ca double-salt materials were also utilized in a combined CO_2_ capture–utilization process aimed at producing syngas [31]. In this process, the CO_2_ sorption capacity was measured at 0.95 mmol/g, with a CO_2_ conversion rate of 65%. However, there are limited studies available on the subject of integrated CO_2_ capture and hydrogenation utilizing alkali-metal-doped Ni/CaO catalysts.

Herein, we successfully synthesized and utilized modified Ni/CaO dual-functional materials for integrated CO_2_ capture (Equation (1)) and reverse water–gas shift (RWGS) reaction (Equation (2)). The RWGS reaction is a highly valuable CO_2_ conversion process that provides a promising opportunity to align the high temperature required for CO_2_ capture with the generation of CO, which can be further utilized as a raw material in various chemical processes to produce valuable chemicals [32]. To achieve this, we employed a facile precipitation–combustion method to synthesize Ni nanoparticles modified with alkali metals (Na, K, and Li) on CaO (M-Ni/CaO), creating effective DFMs for selective CO production through ICCU. Thorough investigations of the physical and chemical properties of the modified Ni/CaO DFMs were conducted using techniques such as XRD, N_2_ adsorption–desorption, SEM, EDX, TEM, CO_2_-TPD, and TG. The introduction of different doping elements (Na, K, and Li) into the Ni/CaO DFMs had a profound impact on CO_2_ capture and CO formation. Additionally, we assessed the cycle stability of the Na-doped Ni/CaO DFMs by performing 20 cycles of integrated CO_2_ capture and hydrogenation. The evaluation results are promising and are expected to provide valuable guidance for the development of highly effective DFMs and the establishment of an efficient pathway to enhance ICCU technology.
(1)$${\mathrm{CaO}+\mathrm{CO}}_{2}\;↔{\;\mathrm{CaCO}}_{3}\;\;\;\;\;\;\;\;\;∆{H\;}_{298K}^{\theta }=-178\;\mathrm{kJ}/\mathrm{mol}$$
(2)$${\mathrm{CaCO}}_{3\;}{+H}_{2}\;↔{\;\mathrm{CaO}+\mathrm{CO}+H}_{2}O\;\;\;\;\;∆{H\;}_{298K}^{\theta }=219\;\mathrm{kJ}/\mathrm{mol}$$

## 2. Experimental Section

### 2.1. Preparation of Ni/CaO and M/CaO (M = Li, Na, and K) DFMs

All reagents were procured from Sinopharm Chemical Reagent Co., Ltd. (Shanghai, China) and utilized without any additional purification steps. All gases were procured from Shanghai Youjiali Liquid Helium Co., Ltd. (Shanghai, China) and had a nominal purity of at least 99.99%. For Li-doped Ni/CaO DFM, 0.70 g of nickel nitrate hexahydrate (Ni(NO_3_)_2_·6H_2_O), 22.14 g of calcium nitrate tetrahydrate (Ca(NO_3_)_2_·4H_2_O), 0.28 g of lithium carbonate (Li_2_CO_3_), and 21.01 g of citric acid monohydrate (C_6_H_8_O_7_·H_2_O) were dissolved in 50 mL of an aqueous solution with a magnetic stirrer at 40 °C under sufficient agitation. Then, 1.0 mol/L of an ammonium carbonate ((NH_4_)_2_CO_3_) solution was added dropwise using a peristaltic pump until the pH value of the mixed solution reached 8.0, where the nickel and calcium were precipitated in the form of carbonates species. The suspension was further evaporated in the superfluous water at 80 °C. The obtained solid was dried at 100 °C overnight and then calcined at 200 °C for 2 h and 700 °C for 5 h, with a heating rate of 2 °C/min. The precursor particles consisted of citric acid complexes and carbonated species of nickel and calcium, which underwent decomposition during the combustion stage at a high calcination temperature. The Na- and K-doped Ni/CaO DFMs were prepared using the same process as described above, and the mass fractions of Ni, the added carbonates, and CaO in each DFM were kept at 2.5 wt%, 5 wt%, and 92.5 wt%, respectively. The obtained sample was denoted as M-Ni/CaO, where M refers to the added alkali metal element (Li, Na, and K). For comparison, the Ni/CaO DFM was also prepared and used as the reference sample.

### 2.2. Characterization of Ni/CaO and M-Ni/CaO (M = Li, Na, and K) DFMs

The crystalline structures of the DFMs were investigated using a Bruker D8 Advance diffractometer to perform X-ray diffraction (XRD) with Cu Kα radiation (λ = 0.15418 nm). The patterns were collected in the 2*θ* range of 10° to 90°with a scanning speed of 8°/min. The actual masses of Ni and alkali metal elements (Li, Na, and K) in the DFMs were measured via inductively coupled plasma−atomic emission spectrometry (ICP−AES) conducted on a PerkinElmer emission spectrometer. The specific surface area and the pore size distribution were determined via N_2_ adsorption–desorption at −196 °C on a Micromeritics ASAP 2020 Sorptometer (Micromeritics Instrument Ltd., Shanghai, China. The DFMs were degassed for 6 h at 200 °C prior to each measurement. The surface morphologies and elemental distribution of the DFMs were performed using an FEI Nova Nano SEM 450 (FEI Company, Hillsboro, OR, USA) scanning electron microscope (SEM) coupled with an EDAX energy-dispersive X-ray spectrometer. All samples were sputtered with a thin layer of Pt via low-vacuum sputter coating before imaging via SEM. The transmission electron microscope (TEM) images of the DFMs were obtained on a JEOL JEM-2010F field-emission scanning electron microscope operating at a working voltage of 200 kV. CO_2_ temperature-programmed desorption (CO_2_-TPD) of the DFMs was determined using an automatic PCA-1200 (Beijing builder Co., Ltd., Beijing, China) chemisorption analyzer. Prior to each measurement, 0.2 g of DFMs was used for pretreatment with Ar at 700 °C for 1 h and then cooled to room temperature. Then, chemical adsorption was conducted using a 10 vol% CO_2_/Ar for 1 h at 80 °C, followed by desorption under pure Ar at temperatures ranging from room temperature to 900 °C.

### 2.3. CO_2_ Capture and Capture–Release of Ni/CaO and M-Ni/CaO (M = Li, Na, and K) DFMs

CO_2_ capture and capture–release experiments of the DFMs were conducted using an SDT650 (TA Instruments, New Castle, DE, USA) thermogravimetric analyzer. For the CO_2_ capture experiment, approximately 8 mg of each DFM was heated to a test temperature (550, 600, 650, or 700 °C) at a heating rate of 10 °C/min in pure Ar at 100 mL/min for 0.5 h to eliminate impurities. Subsequently, a 10 vol% CO_2_/Ar at 100 mL/min was introduced for CO_2_ capture for 2 h. For the CO_2_ capture–release test, about 8 mg of each DFM was heated up to 650 °C in pure Ar for 0.5 h. After 0.5 h of CO_2_ capture in 10 vol% CO_2_/Ar at 100 mL/min, pure Ar was introduced for CO_2_ release for 1.5 h. Additionally, 10 cycles of CO_2_ capture–release tests were performed to evaluate the cyclic stability of the DFMs.

### 2.4. Integrated CO_2_ Capture and Hydrogenation of Ni/CaO and M-Ni/CaO (M = Li, Na, and K) DFMs

The integrated CO_2_ capture and hydrogenation were conducted in a fixed-bed column with a vertical quartz reactor. A thermocouple was put in the middle of the reactor to detect the actual bed temperature. The reactants CO_2_ and H_2_, as well as the dilution gases of Ar, were precisely controlled with mass flow controllers (MFCs) and thoroughly mixed before being introduced into the reactor. Briefly, 1 g of the pelletized sample was pre-reduced in a 10 vol% H_2_/Ar (100 mL/min) at 650 °C for 3 h. Then, CO_2_ capture was conducted in a 10 vol% CO_2_/Ar (100 mL/min) for 1 h. Subsequently, the DFMs were hydrogenated in a 10 vol% H_2_/Ar (100 mL/min) after the reactor was purged with Ar for 10 min. The outlet gas was analyzed online using a GC9800 gas chromatograph. In addition, 20 cycles of combined CO_2_ capture and hydrogenation were performed to investigate the cyclic stability of Na-Ni/CaO.

The concentrations of the reactants (CO_2_ and H_2_) and the product (CO) in the outlet gases were calculated based on Equation (3). In addition, there was no CH_4_ detected in the outlet gases. The conversion of CO_2_ (C_CO_2__) was calculated by following Equation (4).
(3)$$F_{i}\;(\%)=\frac{\left[i\right]}{\left[H_{2}\right]+\left[\mathrm{CO}\right]{+[\mathrm{CO}}_{2}]}\;\times 100\%$$
(4)$$C_{{\mathrm{CO}}_{2}}\;(\%)=\frac{F_{\mathrm{CO}}}{F_{{\mathrm{CO}}_{2}}{+F}_{\mathrm{CO}}}\;\times 100\%$$

An SDT650 (TA Instruments, New Castle, DE, USA) thermogravimetric analyzer was used to analyze the capacities of the captured and released CO_2_ of DFMs. The DFMs before and after the CO_2_ capture stage or the hydrogenation stage were heated to 1000 °C in pure Ar. The CO_2_ capture (N_ac_), release capacity (N_dc_), carbon balance, and CO yield (Y_CO_) of DFMs were calculated based on the following equations:(5)$$N_{\mathrm{ac}}\;\left(\mathrm{mmol}/g\right)=\;\;\frac{{(m}_{0}-{\;m}_{1})\;\times \;1000}{m_{1}{}_{\;}{\;\times \;M}_{{\mathrm{CO}}_{2}}}$$
(6)$$N_{\mathrm{dc}}\;\left(\mathrm{mmol}/g\right){=N}_{\mathrm{ac}}-\frac{{(m}_{2}-{\;m}_{3})\;\times \;1000}{m_{3}{}_{\;}{\;\times \;M}_{{\mathrm{CO}}_{2}}}$$
(7)$$\mathrm{Carbon}\;\mathrm{balance}\;(\%)=\frac{N_{\mathrm{dc}}}{{\;N}_{\mathrm{ac}}}\;\times 100\%$$
(8)$$Y_{\mathrm{CO}}{\;(\mathrm{mmol}/g)=N}_{\mathrm{dc}}{\;\times \;C}_{{\mathrm{CO}}_{2}}$$
where m_0_ and m_1_ represent the masses of the DFMs before and after the CO_2_ capture stage, m_2_ and m_3_ represent the masses of the DFMs before and after the hydrogenation stage, and M_CO_2__ represents the molecular weight of CO_2_.

## 3. Results and Discussion

### 3.1. Characterization of the Ni/CaO and M-Ni/CaO (M = Li, Na, and K) DFMs 

Figure 1a includes the XRD patterns of the calcined Ni/CaO modified with alkali metals (Li-Ni/CaO, Na-Ni/CaO, and K-Ni/CaO) and Ni/CaO. Intense and narrow diffraction peaks at around 32.2°, 37.3°, 53.9°, 64.2°, and 67.4° could be observed for all the prepared materials, which are characteristic of highly crystalline CaO with a cubic structure [33]. The weak peaks around 18.1° and 34.1° are characteristic of Ca(OH)_2_, which is related to the existence of moisture. Furthermore, small diffraction peaks at 43.3° and 62.8° are ascribed to the characteristic peak of NiO. The addition of Na and K did not give rise to new crystalline phases, indicating that alkali metals were present in an amorphous and/or highly dispersed state, and the crystalline structure of the CaO remained unchanged, but a slight decrease in the intensity of the characteristic diffraction peaks of NiO was observed for Na-Ni/CaO and K-Ni/CaO compared with Ni/CaO, which is attributed to the stabilization of metal particles caused by the doping of alkali metals [34,35]. However, a new peak with higher crystallinity attributed to Li_0.3_Ni_1.7_O_2_ was detected for Li-Ni/CaO [36].

The crystal phase of the different samples after subjecting them to the reduction pretreatment at 650 °C in the 10 vol% H_2_/Ar mixture for 3 h was also assessed in the XRD patterns (Figure 1b). All samples maintained the intense and narrow diffraction peaks belonging to CaO, indicating the high stability of this oxide. As expected, no characteristic diffraction peaks were discernible for Ca(OH)_2_, Li_0.3_Ni_1.7_O_2,_ and NiO phases. Diffraction peaks at 44.5° and 51.8° were ascribed to metallic Ni, which originated from the reduction of NiO and/or Li_0.3_Ni_1.7_O_2_. The average crystallite sizes of Ni in the reduced materials were estimated using the Scherrer equation, and the results are shown in Table 1. Na-Ni/CaO and K-Ni/CaO presented relatively smaller Ni crystallites, indicating that the addition of Na and K was conducive to the dispersion of Ni. The small size of Ni particles is considered crucial for achieving high catalytic activity in CO_2_ hydrogenation [37,38]. Therefore, as Li-Ni/CaO had larger Ni crystallites than Ni/CaO, it revealed the most severe Li_0.3_Ni_1.7_O_2_ aggregation.

Regarding textural properties, Figure 2 shows the N_2_ adsorption–desorption isotherms and pore size distributions for the reduced DFMs. All of them exhibited a type-IV isotherm and an H3-shaped hysteresis loop, which are characteristic of slit-shaped pores formed by the decomposition of the citric acid complex and carbonated species. As shown in Table 1, the addition of alkali metals to Ni/CaO resulted in a slight decrease in the specific surface areas to 12.6 and 12.2 m^2^/g for Na-Ni/CaO and K-Ni/CaO, respectively, whereas these DFMs maintained comparable pore volumes as well as average pore sizes. As for Li-Ni/CaO, the specific surface area, pore volume, and average pore size dramatically decreased to 5.9 m^2^/g, 0.04 cm^3^/g, and 19.8 nm, respectively, due to the pore structure blockage caused by the large Li_0.3_Ni_1.7_O_2_ particles. The larger specific surface area of DFMs is believed to correlate with high CO_2_ capture capacity [39]. In addition, Figure 2 shows that Na-Ni/CaO and K-Ni/CaO maintained bimodal pore size distributions with mesopores in the range of 2–5 nm and macropores in the range of 20–100 nm, which was similar to the pore structure of three-dimensional networks consisting of meso- and macropores with excellent elasticity and stability for Ni/CaO, as previously reported [30,31]. Meso- and macropores are responsible for CO_2_ uptake and diffusion control, respectively [40].

To investigate the effect of alkali metal doping on the morphology and particle size of the reduced DFMs, SEM/EDX and TEM were conducted. As shown in Figure 3a–d, the Ni/CaO DFMs prepared via a facile precipitation–combustion method exhibited a three-dimensional network with porous morphology, which was composed of CaO particles ranging in size from 50 to 100 nm. The porous morphology was maintained very well after doping with alkali metals, including Na and K. However, it was noticed that the porous structure of Li-Ni/CaO partly collapsed, which agreed with the obvious decrease in specific surface areas. Additionally, SEM-EDX analysis was conducted to explore the distribution mappings of Na, Ni, and Ca elements in Na-Ni/CaO (Figure 3e,f). As could be observed, Na (blue color), Ni (red color), and Ca (yellow color) elements coexisted with a homogeneous distribution in all the analyzed areas, indicating their good dispersion. From the TEM image of Ni/CaO (Figure 4a), some dark crystal particles were observed with a mean size of ca. 15 nm distributed on the surface, where the lattice distance of 0.203 nm could be clearly seen and represented the lattice plane (111) of the metallic Ni phase [41]. After doping with alkali metals, the average sizes of Ni nanoparticles on the Li-, Na-, and K-Ni/CaO DFMs varied as 21.8 ± 3.8 nm, 12.2 ± 2.0 nm, and 13.3 ± 2.3 nm (Figure 4b–d), respectively, which was consistent with the XRD results (Table 1), suggesting that the introduction of Na or K into Ni/CaO DFMs certainly help to uniform and stabilize Ni particles to achieve the expected catalytic activity.

The effect of alkali metal doping on CO_2_ capture and activation was evaluated in CO_2_-TPD experiments on Ni/CaO and M-Ni/CaO DFMs, which were pre-reduced in situ under H_2_ at 650 °C before CO_2_ adsorption at 80 °C. As shown in Figure 5, three types of CO_2_ desorption peaks were detected for Ni/CaO: (1) the minor peak below 200 °C represents the desorption of the physisorbed CO_2_ (classified as weak basic sites); (2) the peaks located in the temperature range of 250–550 °C typically originate from the chemisorbed CO_2_ on the outer surface, including the Ca(OH)_2_ phase (classified as medium basic sites); (3) the peaks located at temperature between 550 °C and 800 °C are attributed to CO_2_ adsorbed at the bulk part of CaO particles (classified as strong basic sites) [19,42]. In comparison to Ni/CaO, the TPD profiles of Na-Ni/CaO and K-Ni/CaO exhibited similar types of basic sites but with a higher desorption temperature and an increase in peak intensity. As for the Li-Ni/CaO DFM, the low intensity of the desorption peaks suggested a relatively low CO_2_ capture capacity. More specific information about the different basic site distributions was obtained from the integration of the corresponding desorption peaks. As presented in Appendix A, the total basicity of the DFMs follows a descending order of Na-Ni/CaO (987.1 μmol/g) > K-Ni/CaO (879.5 μmol/g) > Ni/CaO (766.7 μmol/g) > Li-Ni/CaO (572.7 μmol/g). Particularly, the number of medium and strong basic sites, which were the main adsorption sites for CO_2_, were significantly increased with Na or K doping, indicating that the deposition of Na or K onto CaO enhanced its affinity for CO_2_ and endowed it with excellent CO_2_ capture capability.

### 3.2. CO_2_ Capture and Capture–Release of Ni/CaO and M-Ni/CaO (M = Li, Na, and K) DFMs

The temperature of CO_2_ capture in the DFMs was tested using the TGA technique under a simulated gas of 10 vol% CO_2_/Ar over a wide temperature range of 50 °C to 1000 °C (Figure 6a). It could be observed that the weight of the as-prepared DFMs remarkably increased in the temperature range of 550–700 °C, in which CO_2_ chemisorption could be activated to allow diffusion processes throughout the bulk of the materials. The carbonated CaO and alkali-metal-modified CaO were completely regenerated, with a sharp decrease in weight occurring at 740–800 °C, which corresponded to the decomposition of CaCO_3_. This suggested that all DFMs exhibited good CO_2_ capture capacities at temperatures between 550 °C and 700 °C. Furthermore, at 550, 600, 650, and 700 °C, the CO_2_ isothermal capture capacities and adsorption rates of all DFMs increased with temperature and reached the maximum level at 650 °C (Appendix A and Appendix A).

Therefore, the isothermal CO_2_ capture–release properties of all DFMs at 650 °C were investigated. As shown in Figure 6b, the capture of CO_2_ on the DFMs followed two different processes. The CO_2_ capture capacity exhibited a rapid increase during the initial stage (within 10 min), which was the chemical-reaction-controlled stage. Subsequently, it slowly increased in the latter stage (within 10–30 min), which was the product-layer-diffusion-controlled stage. Compared with Ni/CaO, the Na- or K-modified DFMs showed a higher CO_2_ capacity both in the reaction-controlled and diffusion-controlled processes, contributing to a higher overall CO_2_ capture capacity. The CO_2_ capture capacities were ranked in the following order: Na-Ni/CaO (11.4 mmol/g) > K-Ni/CaO (10.3 mmol/g) > Ni/CaO (9.9 mmol/g) > Li-Ni/CaO (6.0 mmol/g). In addition, the alkali-metal-modified DFMs showed a shorter time span of CO_2_ release than Ni/CaO, indicating the fast regeneration of the sorbents, which was essential to maintain its structural stability and mitigated CaO sintering [43,44]. The significantly enhanced desorption kinetics of a double-salt adsorbent containing K− and Na−CaO compared with that of bare CaO has previously been reported in several papers [30]. The XRD pattern of Na-Ni/CaO with 15 wt% Na doping after the capture of CO_2_ showed the characteristic peaks of the Na_2_Ca(CO_3_)_2_ phase (Appendix A), which confirmed the formation of double carbonates. The cycle stability of all DFMs was evaluated with 10 cycles of CO_2_ capture and release at 650 °C (Figure 6c). Among them, Na-Ni/CaO showed the best cycle stability, retaining over 90% capacity after 10 cycles with only a slight decrease from 11.4 to 10.3 mmol/g, which was better than Ni/CaO (from 9.9 to 8.4 mmol/g). All these confirmed that not only CO_2_ capture capacities but also the cycle stability of the modified DFMs were improved through the doping of Na and K.

### 3.3. Integrated CO_2_ Capture and Hydrogenation of Ni/CaO and M-Ni/CaO (M = Li, Na, and K) DFMs

The integrated CO_2_ capture and hydrogenation performance of all the DFMs obtained from the fixed bed are illustrated in Figure 7. The whole process includes a CO_2_ capture stage and a hydrogenation stage. In the CO_2_ capture stage, a 10 vol% CO_2_/Ar was introduced into the bed and was maintained for 1 h until the DFMs reached adsorption saturation. Based on the TG profiles of the DFMs after the CO_2_ capture stage (Appendix A), the CO_2_ capture capacities of all DFMs followed an ascending order: Li-Ni/CaO (7.0 mmol/g) < Ni/CaO (11.1 mmol/g) < K-Ni/CaO (11.6 mmol/g) < Na-Ni/CaO (12.0 mmol/g). This trend was consistent with the capture capacities illustrated in Appendix A and the ranking of the number of medium and strong basic sites presented in Appendix A. Following the CO_2_ capture stage, pure Ar was introduced for 10 min to remove the excess CO_2_ in the bed, and then it was switched to 10 vol% H_2_/Ar to initiate the hydrogenation stage. As shown in Table 2, the Ni/CaO DFM achieved a CO_2_ conversion rate and a CO yield of 60.3% and 6.3 mmol/g, respectively, with the hydrogenation stage lasting for 75 min until the complete decomposition of CaCO_3_. It was found that the incorporation of secondary dopants into the NiO–CaO catalysts exerted a significant influence on the overall composition of the product gas, particularly with regard to the water–gas shift reaction [45]. The incorporation of Na and K in the Ni/CaO DFM resulted in higher CO_2_ conversions and CO yields for the doped DFMs compared with Ni/CaO, indicating that the addition of Na and K could promote the hydrogenation of the calcium carbonate species (CaCO_3_ and released CO_2_) towards the production of CO by RWGS. In comparison, the hydrogenation stages of the Na- and K-modified Ni/CaO DFMs were completed in only 45 min, owing to their high CO_2_ conversion and rapid CO_2_ release rates. Among all the DFMs, the Na-Ni/CaO DFM demonstrated the highest CO_2_ conversion rate and CO yield of 62.0% and 7.1 mmol/g, respectively, which is attributed to its abundant basic sites and uniform dispersion of fine Ni particles. The preparation method, reaction types, testing conditions, CO_2_ capture capacities, and product yields in the integrated CO_2_ capture and hydrogenation using different Ni/CaO-based DFMs are listed in Appendix A. The results demonstrated that the Na/Ni–CaO in this work exhibited significantly higher CO_2_ capture capacity and better product yield than most Ni/CaO-based DFMs. Due to the larger Ni particle size and fewer basic sites caused by the poor pore structure, Li-Ni/CaO exhibited the lowest CO_2_ capture capacity (7.0 mmol/g), CO_2_ conversion rate (35.8%), and CO yield (2.3 mol/g) among all the DFMs. However, Li-Ni/CaO also underwent fast hydrogenation, i.e., within 55 min, possibly due to its low CO_2_ capture capacity.

The carbon balance of the DFM was defined as the ratio between the total yields of CO_2_ and CO in the hydrogenation stage and the CO_2_ capture capacity in the capture stage. It was calculated that the carbon balances of all the DFMs were calculated to be higher than 90%, indicating that nearly all of the captured CO_2_ in the capture stage was subsequently released as CO and CO_2_ during the hydrogenation stage.

### 3.4. The Cyclic Integrated CO_2_ Capture and Hydrogenation of Na-Ni/CaO DFMs

The cycle stability of the Na-Ni/CaO DFM was evaluated during 20 cycles of integrated CO_2_ capture and hydrogenation. The change in CO_2_, H_2_, and CO concentrations over time are depicted in Figure 8. The Na-Ni/CaO DFM exhibited similar trends in the concentrations of CO_2_, H_2_, and CO with increasing cycle time, indicating a relatively stable conversion rate of CO_2_. The spent Na-Ni/CaO achieved a CO_2_ conversion rate of 60.5% after 20 cycles. As shown in Appendix A, it was determined that the spent Na-Ni/CaO retained a high CO_2_ capture capacity of 10.8 mmol/g, indicating a mere 10% loss in capture capacity as compared to its initial performance. Figure 9 presents the SEM and TEM images of the spent Na-Ni/CaO after 20 cycles of the integrated process. The images show a slight aggregation of CaO particles and the disappearance of some mesopores, while the three-dimensional network’s porous structure and macropores still remained stable (Figure 9a). Additionally, the porous structure of the spent Na-Ni/CaO was also analyzed using the N_2_ adsorption–desorption isotherms and pore size distribution (Appendix A), which exhibited a slightly lower specific surface area and pore volume than those of fresh Na-Ni/CaO. In addition, the TEM image shows that the spent Na-Ni/CaO also maintained a uniform dispersion of Ni particles, with an average size of 14.5 nm after 20 cycles (Figure 9b). Furthermore, it is noteworthy that neither SEM nor TEM images reveal the existence of carbon deposition, indicating that the decline in CO_2_ conversion could not be attributed to this factor. Therefore, we concluded that the great cycle stability of Na-Ni/CaO might be ascribed to the variation in Na species (Appendix A). One explanation is that Na_2_O forms Na_2_Ca(CO_3_)_2_, which is conducive to CO_2_ capture and release and greatly shortens the cycle time of the whole integrated process. Another possibility is that Na_2_Ca(CO_3_)_2_ can be reduced to Na_2_O, which acts as a high-temperature refractory oxide to retard the sintering of CaO particles during the hydrogenation stage.

## 4. Conclusions

In this study, we successfully synthesized and utilized Ni/CaO DFMs modified with alkaline metals (Li, Na, and K) for producing CO through integrated CO_2_ capture and hydrogenation. The incorporation of Na and K into the DFMs significantly increased the surface basicity, resulting in an enhanced CO_2_ capture capacity. Additionally, Na and K improved the dispersion of Ni particles, leading to smaller particle sizes in the modified DFMs, which in turn contributed to higher catalytic activity. The performance evaluation of the DFMs regarding CO_2_ capture and release revealed that Na- and K-doped Ni/CaO DFMs demonstrated not only high CO_2_ capture capacities but also faster kinetics for both capture and release, surpassing those of Ni/CaO alone. However, the negative effects of Li on the integrated CO_2_ capture and hydrogenation were observed in the Li-Ni/CaO catalyst, which was due to the low surface basicity, pore structure blockage, and large Ni particle sizes. Among all the tested DFMs, Na-Ni/CaO exhibited the highest CO_2_ capture capacity of 12.0 mmol/g and CO_2_ conversion rate of 62.0% with a CO production yield of 7.1 mmol/g in the integrated CO_2_ capture and hydrogenation process at 650 °C. Moreover, the formation of double salts, particularly Na_2_Ca(CO_3_)_2_, played a crucial role in enhancing the CO_2_ capture and release rate of the DFMs, contributing to their stability by preventing the aggregation and sintering of CaO. The recyclability and surface morphology of the spent material clearly indicated that Na-Ni/CaO had high stability even after undergoing 20 cycles.

## Figures and Tables

**Figure 1 materials-16-05430-f001:**
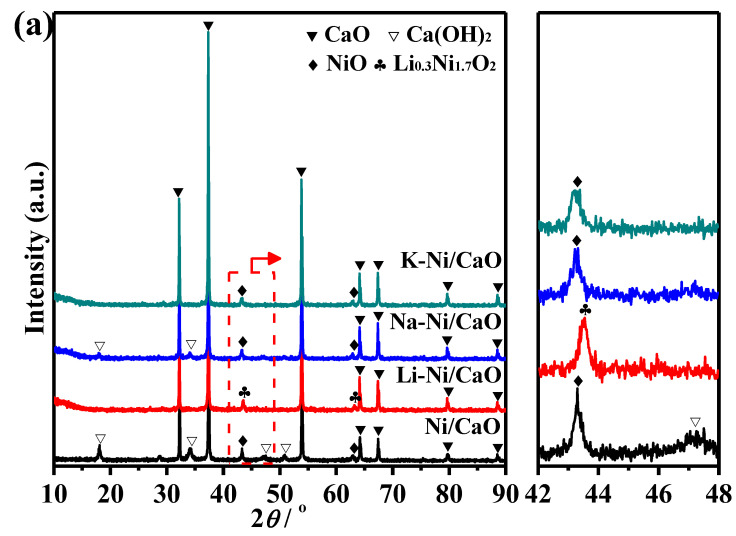
XRD patterns of (**a**) calcined and (**b**) reduced Ni/CaO and M-Ni/CaO (M = Li, Na, and K) DFMs.

**Figure 2 materials-16-05430-f002:**
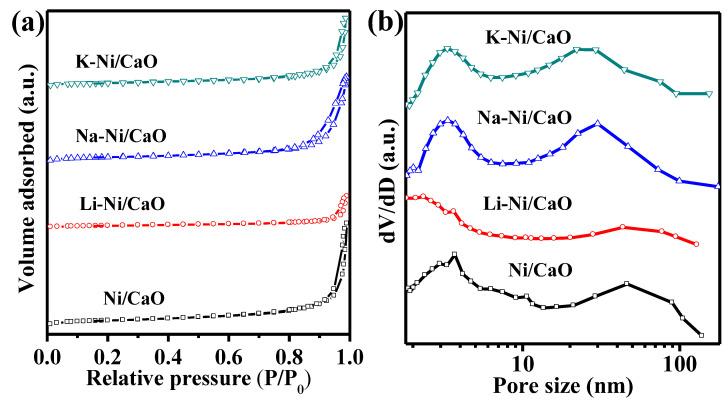
(**a**) N_2_ adsorption–desorption isotherms and (**b**) pore size distributions of the reduced Ni/CaO (☐), Li-Ni/CaO (○), Na-Ni/CaO (△) and K-Ni/CaO (▽) DFMs.

**Figure 3 materials-16-05430-f003:**
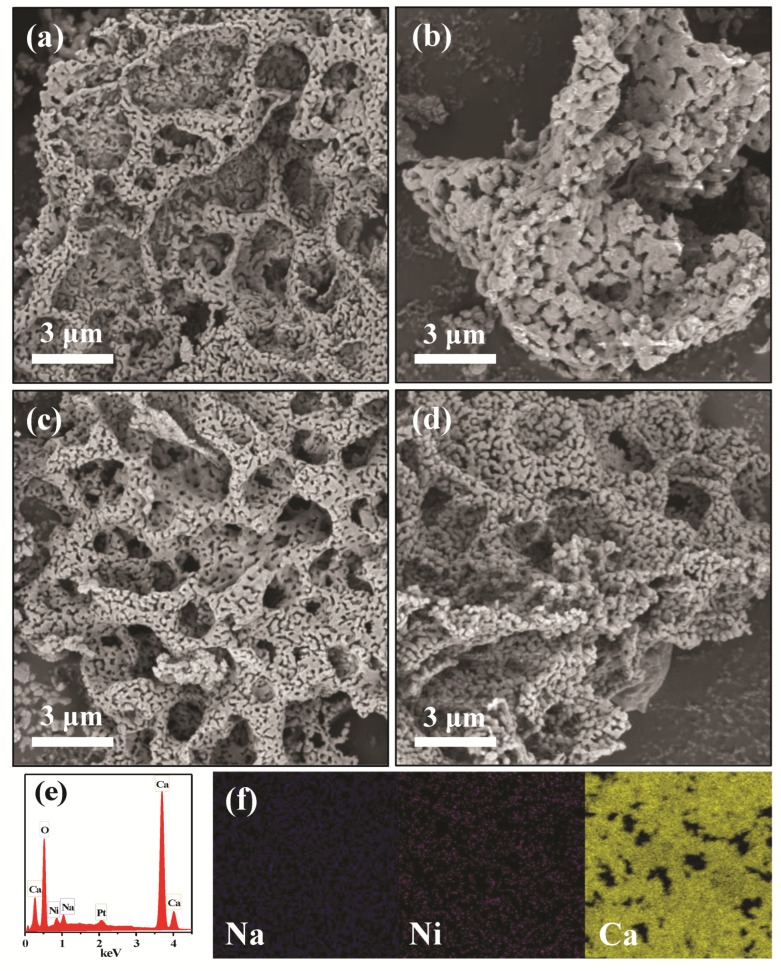
SEM images of (**a**) Ni/CaO, (**b**) Li-Ni/CaO, (**c**) Na-Ni/CaO, (**d**) K-Ni/CaO DFMs, and (**e**) EDX and (**f**) elemental mappings of Na-Ni/CaO DFM.

**Figure 4 materials-16-05430-f004:**
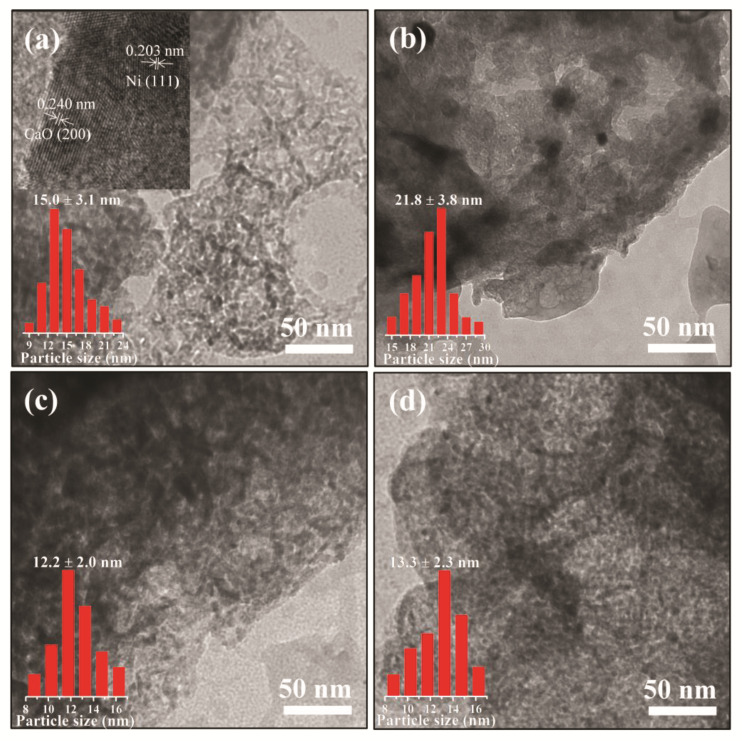
TEM images of (**a**) Ni/CaO, (**b**) Li-Ni/CaO, (**c**) Na-Ni/CaO, and (**d**) K-Ni/CaO DFMs.

**Figure 5 materials-16-05430-f005:**
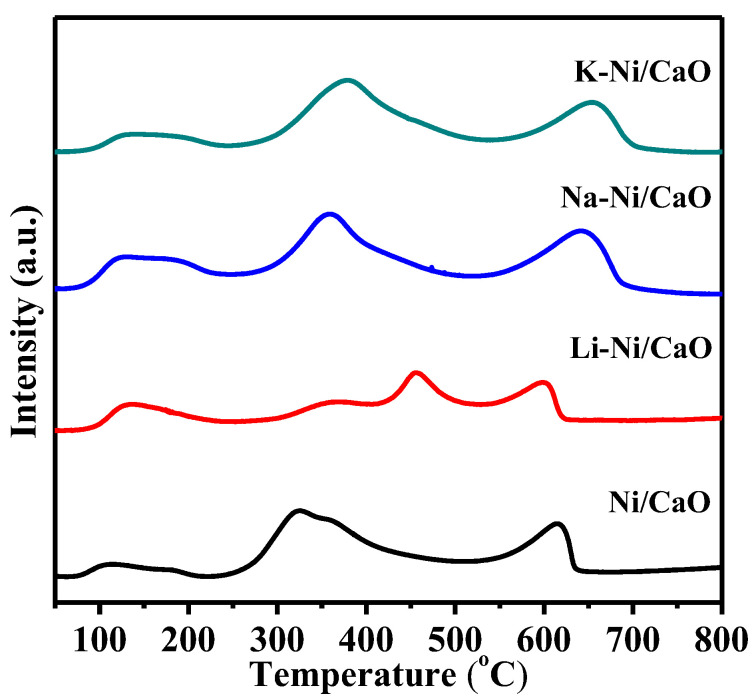
CO_2_-TPD profiles of the reduced Ni/CaO and M-Ni/CaO (M = Li, Na, and K) DFMs.

**Figure 6 materials-16-05430-f006:**
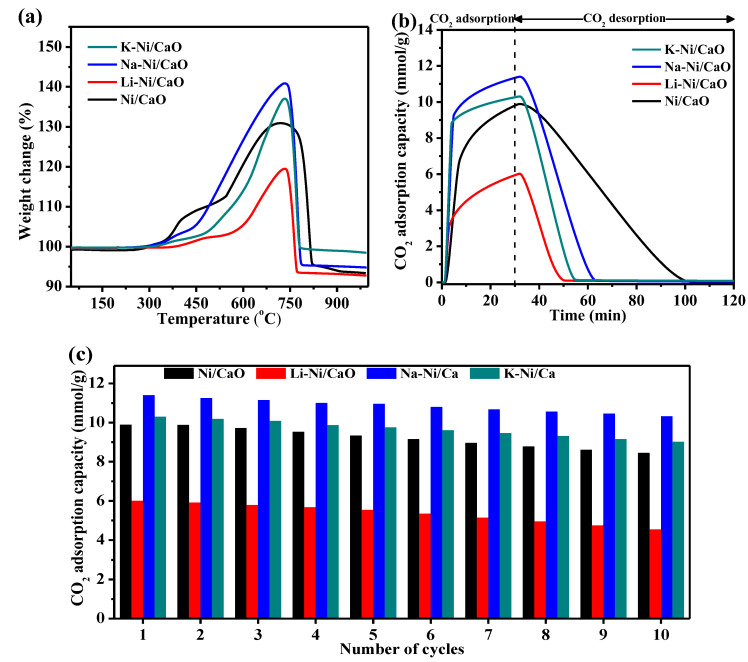
(**a**) The changes in the weights of Ni/CaO and M-Ni/CaO (M = Li, Na, and K) DFMs over the temperature range of 50 °C to 1000 °C; CO_2_ capture–release profiles (**b**) and cyclic CO_2_ capture–release stability (**c**) of Ni/CaO and M-Ni/CaO (M = Li, Na, and K) DFMs at 650 °C.

**Figure 7 materials-16-05430-f007:**
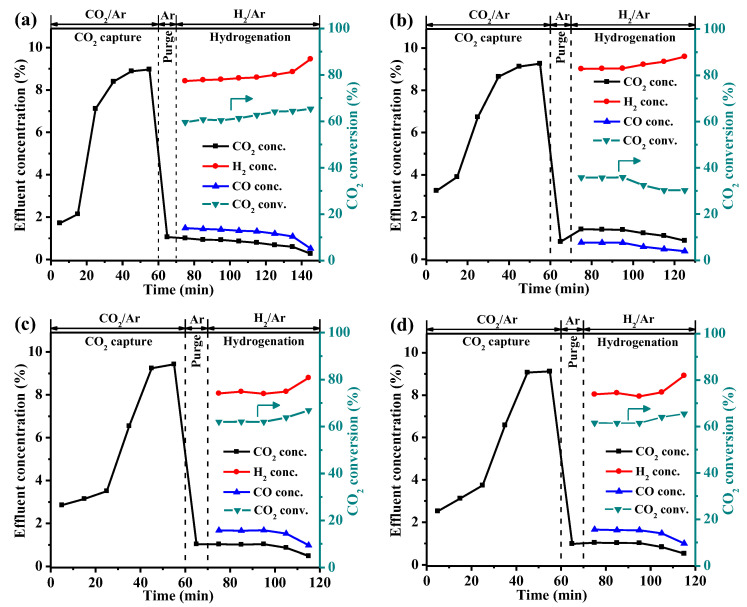
Integrated CO_2_ capture and hydrogenation of (**a**) Ni/CaO ^a^, (**b**) Li-Ni/CaO, (**c**) Na-Ni/CaO, and (**d**) K-Ni/CaO DFM. Temperature: 650 °C; GHSV: 6000 mL·g^−1^·h^−1^. ^a^ Reprinted with permission from ref [28]. Copyright 2023 Elsevier.

**Figure 8 materials-16-05430-f008:**
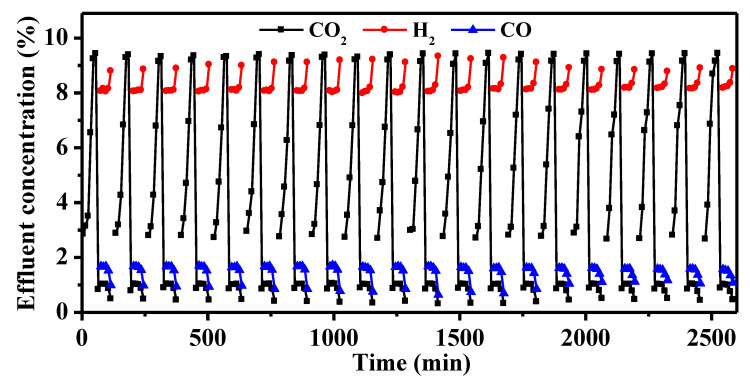
Twenty cycles of integrated CO_2_ capture and hydrogenation of the Na-Ni/CaO DFM. Temperature: 650 °C; GHSV: 6000 mL·g^−1^·h^−1^.

**Figure 9 materials-16-05430-f009:**
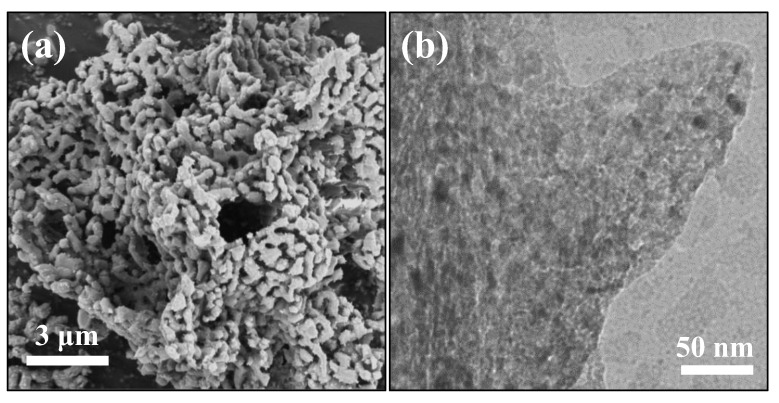
(**a**) SEM and (**b**) TEM images of the spent Na-Ni/CaO DFM after 20 cycles of integrated CO_2_ capture and hydrogenation.

**Table 1 materials-16-05430-t001:** Physicochemical properties of the reduced Ni/CaO and M-Ni/CaO (M = Li, Na, and K) DFMs.

Materials	Ni (wt%) ^a^	Alkali Metal (wt%) ^a^	S_BET_ (m^2^/g)	V_p_ (cm^3^/g)	D_p_ (nm)	Ni Crystallite Size (nm) ^b^	Ni Particle Size (nm) ^c^
Ni/CaO	2.33	N.A.	16.3	0.11	23.7	16.0	15.0 ± 3.1
Li-Ni/CaO	2.31	1.03	5.9	0.04	19.8	22.9	21.8 ± 3.8
Na-Ni/CaO	2.37	2.12	12.6	0.10	23.1	12.9	12.2 ± 2.0
K-Ni/CaO	2.34	2.76	12.2	0.09	22.8	13.7	13.3 ± 2.3

^a^ Determined via ICP analysis. ^b^ Ni crystallite size was calculated using the Scherrer equation. ^c^ Ni particle size was measured using TEM.

**Table 2 materials-16-05430-t002:** The integrated CO_2_ capture and hydrogenation performance of Ni/CaO and M-Ni/CaO (M = Li, Na, and K) DFMs at 650 °C.

Materials	CO_2_ Capture (mmol/g) ^a^	CO_2_ Conversion (%)	CO_2_ Yield (mmol/g)	CO Yield (mmol/g)	Carbon Balance (%) ^b^
Ni/CaO ^c^	11.1	60.3	4.2	6.3	94.4
Li-Ni/CaO	7.0	35.8	4.0	2.3	90.4
Na-Ni/CaO	12.0	62.0	4.4	7.1	95.8
K-Ni/CaO	11.6	61.5	4.2	6.8	95.4

^a^ CO_2_ capture capacities were calculated with Equation (5) and are shown in Appendix A. ^b^ Carbon balances were calculated with Equations (5)–(7) and are shown in Appendix A. ^c^ Reprinted with permission from ref [28]. Copyright 2023 Elsevier.

## Data Availability

No new data were generated.

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
