# Peer review of "The Effect of Alkali Metals (Li, Na, and K) on Ni/CaO Dual-Functional Materials for Integrated CO2 Capture and Hydrogenation"

_materials, 2023, doi:10.3390/ma16155430_

Round 1
Reviewer 1 Report
Unfortunately, I cannot recommend this paper for publication. It needs a serious revision. The topic “Dual-Function Materials for CO2 Capture and Conversion” has been already well studied and constantly developed. Reviews have already been written on this topic. For example, DOI: 10.1021/acs.iecr.0c02218. A lot about NiO/CaO modification methods, about optimal component ratios, about requirements for porous structure, nickel dispersion and thermal stability and etc. has been discussed. But the literature review of this paper contains 25 references, and only four of them are devoted to NiO/CaO [19-22] and two ones describe the improvement of adsorption properties when doping CaO with metals [23,24]. This information does not reflect the research level at all.
1 If the authors want to publish their results, they should rewrite Introduction in more detail so that readers can appreciate the novelty of the results obtained. The results in the work should be compared with the appropriate literary results together with the description of experimental conditions. This is best done as a separate table when discussing the results.
2 The authors wrote that NiO/CaO DFMs were prepared by “a combination of precipitation and combustion method”. In my opinion, this terminology is incorrect. In Section 2.1, the authors should specify when the deposition stage was proceeded and what precipitation product was formed and when the combustion stage was proceeded.
3 Note that the authors do not discuss the literary data on the optimal NiO content and dopants in NiO/CaO DFMs, do not provide the calculated contents, do not confirm the element analysis by any method.
4 Line 110-114. There are no specified pretreatment conditions for the sample in the CO2-TPD experiment.
5 Section 2.3 and 2.4. I won’t go into detail this time, but there is no description of the important details of the experiments, there are repetitions. Please review this text carefully.
6 Section 3.1. Not all PDF cards for crystalline phases are specified when describing the XRD results.
7 Line 178-189. The author wrote that “In contrast, Li-Ni/CaO had larger Ni crystallites compared to Ni/CaO, corresponding to the most severe Li0.3Ni1.7O2 aggregation». But, in Table 1, Li-Ni/CaO had smaller Ni crystallites.
8 Line 193. The author wrote that “caused by the large Li0.3Ni1.7O2 particles.” The particle size of Li0.3Ni1.7O2, its properties and effect mechanism are not discussed in the paper. The literature data should be pointed out at least.
9 Again, the data on adsorption and catalytic conversion should be compared with the literature data.
10 Table 2. What is the accuracy of the experimental data? Are the values on CO2 capture (11.1 and 12.0), CO2 conversion (60.3 and 62), CO2 yield (4.2 and 4.4) so different?
11. Conclusion. The phrase “a novel integrated CO2 capture and hydrogenation process” should be supported by the comparison with literature data. How new is the statement “Moreover, the formation of double salts, particularly Na2Ca(CO3)2, played a crucial role in enhancing the CO2 capture and release rate of DFMs” and etc.? I recommend that the conclusions be rewritten in accordance with the above remarks.
Author Response
Reviewer 1: Unfortunately, I cannot recommend this paper for publication. It needs a serious revision. The topic “Dual-Function Materials for CO2 Capture and Conversion” has been already well studied and constantly developed. Reviews have already been written on this topic. For example, DOI: 10.1021/acs.iecr.0c02218. A lot about NiO/CaO modification methods, about optimal component ratios, about requirements for porous structure, nickel dispersion and thermal stability and etc. has been discussed. But the literature review of this paper contains 25 references, and only four of them are devoted to NiO/CaO [19-22] and two ones describe the improvement of adsorption properties when doping CaO with metals [23,24]. This information does not reflect the research level at all.
Response: We are very grateful for the reviewer’s positive comments and helpful suggestions. Those comments are all highly valuable and very helpful for revising and improving our paper.
- If the authors want to publish their results, they should rewrite Introduction in more detail so that readers can appreciate the novelty of the results obtained. The results in the work should be compared with the appropriate literary results together with the description of experimental conditions. This is best done as a separate table when discussing the results.
Response: Thank you for the reviewer’s comments. As the reviewer’s suggestions, we have rewritten the introduction in more detail and added some research work of the literature in the introduction. In addition, the combined CO2 capture and hydrogenation performance of some other Ni/CaO based DFMs have been also compared with the Na-Ni/CaO DFM in this work, as illustrated in Table R1. The corresponding supplements have been added in the revised supplementary materials as Table S3 on Page S5.
Table R1. The combined CO2 capture and hydrogenation performance over our proposed Na-Ni/CaO DFM and recently documented Ni/CaO based DFMs.
|
DFMs |
Preparation method |
Reaction |
Temperature (oC) |
Feed composition CO2 capture + conversion |
CO2 capacity (mmol/g) |
Product yield (mmol/g) |
Ref. |
|
Fe5Co5Mg10CaO |
One-pot sol-gel |
RWGS |
650 |
10% CO2 + 100% H2 |
9.2 |
8.28 |
1 |
|
15NiCa |
Wet impregnation |
Methanation |
520 |
10% CO2 + 10% H2 |
- |
0.14 |
2 |
|
15NiNa |
- |
0.19 |
|||||
|
Ca1Ni0.1 |
Sol-gel |
RWGS |
650 |
15% CO2 + 5% H2 |
15.0 |
6.9 |
3 |
|
Ca1Ni0.1Ce0.033 |
14.1 |
7.3 |
|||||
|
Ni/CaO |
|
Methanation |
500 |
10% CO2/10% H2O + 10% H2 |
8.96 |
8.34 |
4 |
|
Ni-CaO/γ-Al2O3 |
impregation |
Methanation |
320 |
9.5% CO2 + 10% H2 |
0.31 |
0.14 |
5 |
|
1%NiCaO |
One-pot |
Methanation |
550 |
15% CO2 + 100% |
9.2 |
2.0 |
6 |
|
1%Ni/CeO2-CaO-phy |
physical mixing |
15.3 |
8.0 |
||||
|
Ni/CS-P30-C |
Sol-gel |
RWGS |
650 |
10% CO2 + 5% H2 |
13.86 |
5.52 |
7 |
|
Ni/CS-P30-C-P |
8.95 |
5.33 |
|||||
|
Ni/CeO2-CaO |
physical mixture |
RWGS |
650 |
10% CO2 + 5% H2 |
9.6 |
2.7 |
8 |
|
Ni/CaO |
Precipitation-combustion |
RWGS |
650 |
10% CO2 + 10% H2 |
11.1 |
6.3 |
This work |
|
Na-Ni/CaO |
12.0 |
7.0 |
- The authors wrote that NiO/CaO DFMs were prepared by “a combination of precipitation and combustion method”. In my opinion, this terminology is incorrect. In Section 2.1, the authors should specify when the deposition stage was proceeded and what precipitation product was formed and when the combustion stage was proceeded.
Response: Thanks a lot for the reviewer’s comments. As the reviewer’s suggestions, we have added “where the nickel and calcium were precipitated in the form of carbonates species.” (Page 3, Line 97-98) and “The precursor particles consisted of citric acid complexes and carbonated species of nickel and calcium underwent decomposition during the combustion stage at high calcination temperature.” (Page 3, Line 100-102) in the revised manuscript to describe when the deposition stage was proceeded and what precipitation product was formed and when the combustion stage was proceeded.
- Note that the authors do not discuss the literary data on the optimal NiO content and dopants in NiO/CaO DFMs, do not provide the calculated contents, do not confirm the element analysis by any method.
Response: Thanks you for the reviewer’s comments. We apologize for the omission of discussing the impacts of Ni and dopant content in this paper, as these aspects will be the primary focus of our forthcoming research. As the reviewer’s suggestions, we have added “and the mass fractions of Ni, added carbonates and CaO in each DFM were kept at 2.5 wt%,5 wt% and 92.5 wt%, respectively.” (Page 3, Line 104-106) in the revised manuscript. Meanwhile, the actual Ni and alkali metal (Li, Na, K) mass fractions in Ni/CaO and M-Ni/CaO (Li, Na, K) DFMs shown in Table 1 was determined by ICP analysis. (Page 6, Line 217-219)
- Line 110-114. There are no specified pretreatment conditions for the sample in the CO2-TPD experiment.
Response: Thanks a lot for the reviewer’s suggestions. We have added the “Prior to each measurement, 0.2 g of DFMs underwent pretreatment with Ar at 700 °C for 1 h and then cooled to room temperature.” (Page 3, Line 124-125) in the revised manuscript.
- Section 2.3 and 2.4. I won’t go into detail this time, but there is no description of the important details of the experiments, there are repetitions. Please review this text carefully.
Response: Thank you for the reviewer’s suggestions. We have added the “For the CO2 capture experiment, approximately 8 mg of DFMs were heated to a test temperature (550, 600, 650 or 700 °C) at a heating rate of 10 °C /min in pure Ar at 100 mL/min for 0.5 h to eliminate impurities.” (Page 3, Line 130-132), “Additionally, 10 cycles of CO2 capture-release test were performed to evaluated the cyclic stability of the DFMs.” (Page 3, Line 136-137) and “In addition, a 20 cycles of the combined CO2 capture and hydrogenation was performed to investigate the cyclic stability of the Na-Ni/CaO.” (Page 4, Line 148-149) in the revised manuscript.
- Section 3.1. Not all PDF cards for crystalline phases are specified when describing the XRD results.
Response: Thanks you for the reviewer’s comments. We have revised the relevant description of PDF cards (Page 4, Line 172-173) in the revised manuscript.
- Line 178-189. The author wrote that “In contrast, Li-Ni/CaO had larger Ni crystallites compared to Ni/CaO, corresponding to the most severe Li0.3Ni1.7O2 aggregation. But, in Table 1, Li-Ni/CaO had smaller Ni crystallites.
Response: Thanks you for the reviewer’s suggestions. We are sorry that we filled in the incorrect position of Ni crystallites sizes of the M-Ni/CaO (Li, Na, K) in Table 1. We recalculated the Ni crystallites size by Scherrer equation and verified that the Ni crystallites sizes of the Li-Ni/CaO, Na-Ni/CaO and K-Ni/CaO were 22.9 nm, 12.9 nm and 13.7 nm, respectively. And we have corrected the Ni crystallites sizes of the M-Ni/CaO (Li, Na, K) in Table 1 (Page 6, Line 217-219) in the revised manuscript.
- Line 193. The author wrote that “caused by the large Li0.3Ni1.7O2 particles.” The particle size of Li0.3Ni1.7O2, its properties and effect mechanism are not discussed in the paper. The literature data should be pointed out at least.
Response: Thanks you for the reviewer’s comments. It was demonstrated that the addition of Li in the Ni/MgO catalyst will cause the intensive growth in the mean Ni particles size (Appl. Catal., A 1999, 187, 127–140). We assumed that the introduction of Li would lead to the formation of large particles of Li0.3Ni1.7O2, resulting in an increase in the size of Ni particles after reduction. As the reviewe’s suggested, we have cited the reference (Page 4, Line 182) in the revised manuscript.
- Again, the data on adsorption and catalytic conversion should be compared with the literature data.
Response: Thanks you for the reviewer’s comments. As the reviewer’s suggestion, the combined CO2 capture and hydrogenation performance of some other Ni/CaO based DFMs have been also compared with the Na-Ni/CaO DFM in this work, as illustrated in Table R1. The corresponding supplements have been added in the revised supplementary materials as Table S3 on Page S5.
- Table 2. What is the accuracy of the experimental data? Are the values on CO2 capture (11.1 and 12.0), CO2 conversion (60.3 and 62), CO2 yield (4.2 and 4.4) so different?
Response: Thanks a lot for the reviewer’s comments. We confirmed that the experimental data include the CO2 capture capacities,CO2 conversions and CO2 yields of all DFMs in Table 2 were accurate. For the Ni/CaO and Na-Ni/CaO, the CO2 capture capacity were 11.1 mmol/g and 12.0 mmol/g, respectively, which were calculated according Eq. (6) by Figure S3a. And the carbon balances were 94.4% and 95.8%, respectively, which were calculated according Eq. (6), (7), (8) by Figure S3a and b. The CO2 conversions of the Ni/CaO and Na-Ni/CaO were 60.3% and 62%, respectively, which were obtained by the Figure 7a and c. And the CO2 yield were 4.2 mmol/g and 4.4 mmol/g, which were calculated by the CO2 capture capacity×carbon balance×(1-CO2 conversion).
- Conclusion. The phrase “a novel integrated CO2 capture and hydrogenation process” should be supported by the comparison with literature data. How new is the statement “Moreover, the formation of double salts, particularly Na2Ca(CO3)2, played a crucial role in enhancing the CO2capture and release rate of DFMs” and etc.? I recommend that the conclusions be rewritten in accordance with the above remarks.
Response: Thank you for the reviewer’s comments. As the reviewer’s suggestion, the combined CO2 capture and hydrogenation performance of some other Ni/CaO based DFMs have been also compared with the Na-Ni/Ca DFM in this work, as illustrated in Table R1 and Table S3 in the revised supplementary materials. Meanwhile, we have revised the conclusions (Page 13, Line 392-410) in the revised manuscript.

Reviewer 2 Report
This manuscript describes an experimental study of nickel-calcium oxide materials treated with alkali metals evaluated for both CO2 capture and subsequent hydrogenation. The work provides interesting results and analysis, but there are several issues that must be corrected prior to publication, some of which are fairly serious. However, I am still recommending minor revision due to the overall apparent quality.
In Sections 2.2 and 2.3, please report the manufacturer/supplier and purities of the reactant gases used.
What was the thermal treatment in Ar (described in line 118)? The temperature and ramp rate should be included in the text. This is perhaps described in lines 120-121, but it is not clear. If this is the same thermal treatment, the text can be written more concisely to eliminate the redundant description.
Equations 3, 4, and 5 include CH4, but no methane formation is reported in the results or discussed in the manuscript. Was no methane formed? If so, the equations (and text description) should be rewritten to exclude it (and state that methane was not detected). If it was formed, the amount must be reported in the results and discussed.
m3 is not identified in Equation 7. If it is a mass, it would probably be better to rename w1, w2, and w3 as “masses” (m1, m2, and m3) because that is a more correct description than “weight”. Regardless, the existing m3 must be defined in the text (or if it is incorrectly written and is actually w3, this correction must be made).
The authors’ discussion of the Ni crystallite and particle sizes shown in Table 1 appears to be inconsistent and/or incorrect. Lines 175-179 state “Na-Ni/CaO and K-Ni/CaO presented the relatively smaller Ni crystallites, indicating that the addition of Na and K was conducive to the dispersion of Ni. The small size of Ni particles is considered crucial for achieving high catalytic activity in CO2 hydrogenation [28,29]. In contrast, Li-Ni/CaO had larger Ni crystallites compared to Ni/CaO, corresponding to the most severe Li0.3Ni1.7O2 aggregation.” However, the lithium material has the smallest value in the “Ni crystallite size” column in Table 1, so what the authors have written in the text is not correct. Then, in lines 219-221, the authors state, “After doping with alkali metals, the average sizes of Ni nanoparticles on the Li-, Na-, and K-Ni/CaO DFMs varied as 21.8 ± 3.8 nm, 12.2 ± 2.0 nm, and 13.3 ± 2.3 nm, respectively (Figure 4b-d), which was consistent with the XRD results (Table 1).” These TEM results are not consistent with the XRD results, which showed that the lithium had the smallest Ni crystallites. The authors need to correct their statements and/or the data in Table 1 so that this discussion is actually consistent and makes sense. The two columns labeled “Ni crystallite size” and “Ni particle size” are not at all “consistent” and this discrepancy must be reconciled. Further, the difference between these data columns should be noted in the table, perhaps as footnotes, stating that one is from XRD using the Scherrer equation and the other is from TEM using optical image processing (if that is how they were determined).
The Figure 2 caption needs to include the (a) and (b) descriptor labels.
Why is there a platinum (Pt) peak in Figure 3(e) and no nickel? The authors need to explain this in the text. Or is this actually nickel (Ni)? If so, the authors have been careless in preparing their manuscript and must correct this serious error.
The wording in lines 294-295 is unclear and makes it sound as if there are two stages of capture and reaction in each stage (“The whole process includes two stages of CO2 capture stage and the hydrogenation stage.”). A less-confusing description might be simply “The whole process includes a CO2 capture stage and a hydrogenation stage.”
Minor typographical issues (there are many others; these are just a few that I noted):
Line 43: “forms” should be “form”
Line 52: “carry” should be “are carried”
Line 53: “their” should be “its”
Line 158: “exist” should be “existence”
Line 199: “previous” should be “previously”
Line 352: “conductive” should be “conducive”
This manuscript describes an experimental study of nickel-calcium oxide materials treated with alkali metals evaluated for both CO2 capture and subsequent hydrogenation. The work provides interesting results and analysis, but there are several issues that must be corrected prior to publication, some of which are fairly serious. However, I am still recommending minor revision due to the overall apparent quality.
In Sections 2.2 and 2.3, please report the manufacturer/supplier and purities of the reactant gases used.
What was the thermal treatment in Ar (described in line 118)? The temperature and ramp rate should be included in the text. This is perhaps described in lines 120-121, but it is not clear. If this is the same thermal treatment, the text can be written more concisely to eliminate the redundant description.
Equations 3, 4, and 5 include CH4, but no methane formation is reported in the results or discussed in the manuscript. Was no methane formed? If so, the equations (and text description) should be rewritten to exclude it (and state that methane was not detected). If it was formed, the amount must be reported in the results and discussed.
m3 is not identified in Equation 7. If it is a mass, it would probably be better to rename w1, w2, and w3 as “masses” (m1, m2, and m3) because that is a more correct description than “weight”. Regardless, the existing m3 must be defined in the text (or if it is incorrectly written and is actually w3, this correction must be made).
The authors’ discussion of the Ni crystallite and particle sizes shown in Table 1 appears to be inconsistent and/or incorrect. Lines 175-179 state “Na-Ni/CaO and K-Ni/CaO presented the relatively smaller Ni crystallites, indicating that the addition of Na and K was conducive to the dispersion of Ni. The small size of Ni particles is considered crucial for achieving high catalytic activity in CO2 hydrogenation [28,29]. In contrast, Li-Ni/CaO had larger Ni crystallites compared to Ni/CaO, corresponding to the most severe Li0.3Ni1.7O2 aggregation.” However, the lithium material has the smallest value in the “Ni crystallite size” column in Table 1, so what the authors have written in the text is not correct. Then, in lines 219-221, the authors state, “After doping with alkali metals, the average sizes of Ni nanoparticles on the Li-, Na-, and K-Ni/CaO DFMs varied as 21.8 ± 3.8 nm, 12.2 ± 2.0 nm, and 13.3 ± 2.3 nm, respectively (Figure 4b-d), which was consistent with the XRD results (Table 1).” These TEM results are not consistent with the XRD results, which showed that the lithium had the smallest Ni crystallites. The authors need to correct their statements and/or the data in Table 1 so that this discussion is actually consistent and makes sense. The two columns labeled “Ni crystallite size” and “Ni particle size” are not at all “consistent” and this discrepancy must be reconciled. Further, the difference between these data columns should be noted in the table, perhaps as footnotes, stating that one is from XRD using the Scherrer equation and the other is from TEM using optical image processing (if that is how they were determined).
The Figure 2 caption needs to include the (a) and (b) descriptor labels.
Why is there a platinum (Pt) peak in Figure 3(e) and no nickel? The authors need to explain this in the text. Or is this actually nickel (Ni)? If so, the authors have been careless in preparing their manuscript and must correct this serious error.
The wording in lines 294-295 is unclear and makes it sound as if there are two stages of capture and reaction in each stage (“The whole process includes two stages of CO2 capture stage and the hydrogenation stage.”). A less-confusing description might be simply “The whole process includes a CO2 capture stage and a hydrogenation stage.”
Minor typographical issues (there are many others; these are just a few that I noted):
Line 43: “forms” should be “form”
Line 52: “carry” should be “are carried”
Line 53: “their” should be “its”
Line 158: “exist” should be “existence”
Line 199: “previous” should be “previously”
Line 352: “conductive” should be “conducive”
The authors' intended meaning is generally clear, but the English quality could be polished throughout. An inexhaustive list of minor improvements is included at the end of my previous comments.
Author Response
Reviewer 2: This manuscript describes an experimental study of nickel-calcium oxide materials treated with alkali metals evaluated for both CO2 capture and subsequent hydrogenation. The work provides interesting results and analysis, but there are several issues that must be corrected prior to publication, some of which are fairly serious. However, I am still recommending minor revision due to the overall apparent quality.
Response: We are very grateful for the reviewer’s positive comments and helpful suggestions. Those comments are all highly valuable and very helpful for revising and improving our paper.
In Sections 2.2 and 2.3, please report the manufacturer/supplier and purities of the reactant gases used.
Response: Thanks a lot for the reviewer’s comments. We have added “All gases were procured from Shanghai Youjiali Liquid Helium Co., Ltd and had a nominal purity of at least 99.99%.”(Page 2, Line 90-91) in the revised manuscript.
What was the thermal treatment in Ar (described in line 118)? The temperature and ramp rate should be included in the text. This is perhaps described in lines 120-121, but it is not clear. If this is the same thermal treatment, the text can be written more concisely to eliminate the redundant description.
Response: Thank you for the reviewer’s suggestions. We have corrected “For the CO2 capture experiment, approximately 8 mg of DFMs were subjected to a thermal treatment in pure Ar for 0.5 h to eliminate impurities.” to “For the CO2 capture experiment, approximately 8 mg of DFMs were heated to a test temperature (550, 600, 650 or 700 °C) at a heating rate of 10 °C /min in pure Ar at 100 mL/min for 0.5 h to eliminate impurities.” (Page 3, Line 130-132) in the revised manuscript.
Equations 3, 4, and 5 include CH4, but no methane formation is reported in the results or discussed in the manuscript. Was no methane formed? If so, the equations (and text description) should be rewritten to exclude it (and state that methane was not detected). If it was formed, the amount must be reported in the results and discussed.
Response: Thanks a lot for the reviewer’s suggestions. In fact, there was no methane formed in the combined CO2 capture and hydrogenation at 650 oC. As the reviewer’s suggestion, we have added “In addition, there was no CH4 detected in the outlet gases.”, revised the Equations 3, 4, and deleted the Equations 5 (Page 4, Lines 151-152) in the revised manuscript.
m3 is not identified in Equation 7. If it is a mass, it would probably be better to rename w1, w2, and w3 as “masses” (m1, m2, and m3) because that is a more correct description than “weight”. Regardless, the existing m3 must be defined in the text (or if it is incorrectly written and is actually w3, this correction must be made).
Response: Thank you for the reviewer’s comments. As the reviewer’s suggestion, we have corrected the w0, w1, w2, and w3 to m0, m1, m2 and m3 in the Equation 6 and 7. In addition, we have corrected “where w0 and w1 represent the weights of the DFMs before and after the CO2 capture stage, w2 and w3 represent the weights of the DFMs before and after the hydrogenation stage,” to “the where m0 and m1 represent the masses of the DFMs before and after the CO2 capture stage, m2 and m3 represent the masses of the DFMs before and after the hydrogenation stage,” (Page 4, Lines 160-166) in the revised manuscript.
The authors’ discussion of the Ni crystallite and particle sizes shown in Table 1 appears to be inconsistent and/or incorrect. Lines 175-179 state “Na-Ni/CaO and K-Ni/CaO presented the relatively smaller Ni crystallites, indicating that the addition of Na and K was conducive to the dispersion of Ni. The small size of Ni particles is considered crucial for achieving high catalytic activity in CO2 hydrogenation [28,29]. In contrast, Li-Ni/CaO had larger Ni crystallites compared to Ni/CaO, corresponding to the most severe Li0.3Ni1.7O2 aggregation.” However, the lithium material has the smallest value in the “Ni crystallite size” column in Table 1, so what the authors have written in the text is not correct. Then, in lines 219-221, the authors state, “After doping with alkali metals, the average sizes of Ni nanoparticles on the Li-, Na-, and K-Ni/CaO DFMs varied as 21.8 ± 3.8 nm, 12.2 ± 2.0 nm, and 13.3 ± 2.3 nm, respectively (Figure 4b-d), which was consistent with the XRD results (Table 1).” These TEM results are not consistent with the XRD results, which showed that the lithium had the smallest Ni crystallites. The authors need to correct their statements and/or the data in Table 1 so that this discussion is actually consistent and makes sense. The two columns labeled “Ni crystallite size” and “Ni particle size” are not at all “consistent” and this discrepancy must be reconciled. Further, the difference between these data columns should be noted in the table, perhaps as footnotes, stating that one is from XRD using the Scherrer equation and the other is from TEM using optical image processing (if that is how they were determined).
Response: Thanks a lot for the reviewer’s comments. We are sorry that we filled in the incorrect position of Ni crystallites sizes of the M-Ni/CaO (Li, Na, K) in Table 1. We recalculated the Ni crystallites size by Scherrer equation and verified that the Ni crystallites sizes of the Li-Ni/CaO, Na-Ni/CaO and K-Ni/CaO were 22.9 nm, 12.9 nm and 13.7 nm, respectively. And we have corrected the Ni crystallites sizes of the M-Ni/CaO (Li, Na, K) and added the footnotes of “b Ni crystallite size was calculated by Scherrer equation. c Ni particle size was measured by TEM.” in Table 1 (Page 6, Line 217-219) in the revised manuscript.
The Figure 2 caption needs to include the (a) and (b) descriptor labels.
Response: Thank you for the reviewer’s comments. We have added the (a) and (b) descriptor labels in the caption of Figure 2 (Page 6, Lines 221) in the revised manuscript.
Why is there a platinum (Pt) peak in Figure 3(e) and no nickel? The authors need to explain this in the text. Or is this actually nickel (Ni)? If so, the authors have been careless in preparing their manuscript and must correct this serious error.
Response: Thanks a lot for the reviewer’s valuable comments. The Ni peak in Figure 3(e) was observed at 0.85 eV. In fact, all samples were sputtered with a thin layer of Pt by low vacuum sputter coating before imaging by SEM. The Pt element could be detected by EDX and a Pt peak appeared in the EDX spectrum. We added the “All samples were sputtered with a thin layer of Pt by low vacuum sputter coating before imaging by SEM” (Page 3, Lines 119-120) in the revised manuscript.
The wording in lines 294-295 is unclear and makes it sound as if there are two stages of capture and reaction in each stage (“The whole process includes two stages of CO2 capture stage and the hydrogenation stage.”). A less-confusing description might be simply “The whole process includes a CO2 capture stage and a hydrogenation stage.”
Response: Thanks you for the reviewer’s comments. We have corrected the “The whole process includes two stages of CO2 capture stage and the hydrogenation stage.” into the “The whole process includes a CO2 capture stage and a hydrogenation stage.” (Page 10, Lines 312-313) in the revised manuscript.
Minor typographical issues (there are many others; these are just a few that I noted):
Line 43: “forms” should be “form”
Line 52: “carry” should be “are carried”
Line 53: “their” should be “its”
Line 158: “exist” should be “existence”
Line 199: “previous” should be “previously”
Line 352: “conductive” should be “conducive”
Response: Thanks a lot for the reviewer’s valuable suggestions. We have rectified the aforementioned grammatical errors and conducted a thorough review of the entire paper, resulting in further corrections.

Reviewer 3 Report
Title: Effect of alkali metal (Li, Na, K,) on Ni/CaO dual functional materials for integrated CO2 capture and hydrogenation
After a careful reading of this manuscript I think it can be published after minor revision.
In the Introduction:
authors wrote “It is concluded that a series of double salts such as K- and 61 Na- promoted Ca adsorbents, have enhanced the CO2 capture capacity and cycle stability 62 of CaO” – have Na or similar metals been already used for modify Ni/CaO? If yes, specify the differences/novelty of this work compared with already reported researches.
In the Results and discussion:
Have you an idea about why the addition of Li decreased the CO2 capture capacity and also its conversion? you may add this explanation in the manuscript.
Author Response
Reviewer 3: Title: Effect of alkali metal (Li, Na, K,) on Ni/CaO dual functional materials for integrated CO2 capture and hydrogenation
After a careful reading of this manuscript I think it can be published after minor revision.
Response: We are very grateful for the reviewer’s positive comments and helpful suggestions.
In the Introduction:
authors wrote “It is concluded that a series of double salts such as K- and 61 Na- promoted Ca adsorbents, have enhanced the CO2 capture capacity and cycle stability 62 of CaO” – have Na or similar metals been already used for modify Ni/CaO? If yes, specify the differences/novelty of this work compared with already reported researches.
Response: Thanks a lot for the reviewer’s comments. For the Ni20@(K−Ca)50/(γ-Al2O3)50 and Ni20@(Na−Ca)50/(γ-Al2O3)50 used in reference 31, however, the Ni-impregnated CaO- and MgO-based double salts were supported on γ-Al2O3, resulting in a limited amount of CaO available for CO2 capture. Consequently, the CO2 capture capacities were only 0.63 and 0.99 mol/g, respectively, which was much lower than that of Ni/CaO and M-Ni/CaO (Li, Na, K) in our paper. In addition, the two DMFs were employed for in situ capture and utilization of CO2 in syngas production from dry reforming of ethane (DRE), which was different from the integrated CO2 capture and hydrogenation by RWGS as described in our paper. At last, the stability of the two DMFs during cyclic reactions remained to be evaluated. In addition, we have added “Ahmed et al. conducted a study on a series of double salts, which included K- and Na-promoted Ca adsorbents, specifically designed for CO2 capture at 650 °C. The results of their research showed that the K−Ca double salt displayed an exceptional CO2 adsorption capacity of 10.7 mmol/g [30]. The K−Ca double salt materials were also utilized in a combined CO2 capture-utilization process aimed at producing syngas [31]. In this process, the CO2 sorption capacity was measured at 0.95 mmol/g, with a CO2 conversion rate of 65%. However, there is limited literature available on the subject of integrated CO2 capture and hydrogenation utilizing alkali metal-doped Ni/CaO cata-lysts.” (Page 2, Lines 60-67) in the revised manuscript.
In the Results and discussion:
Have you an idea about why the addition of Li decreased the CO2 capture capacity and also its conversion? you may add this explanation in the manuscript.
Response: Thanks a lot for the reviewer’s valuable suggestions. We have added the “Due to the larger Ni particle size and fewer basic sites cause by the poor pore structure, the Li-Ni/CaO exhibited the lowest CO2 capture capacity (7.0 mmol/g), CO2 conversion (35.8%) and CO yield (2.3 mol/g) in all DFMs.” (Page 11, Lines 339-342) in the revised manuscript.

Round 2
Reviewer 1 Report
My respect to authors. They answered all my questions. The paper may be recommended to be publish.